# Efficient Upper Limb Position Estimation Based on Angular Displacement Sensors for Wearable Devices

**DOI:** 10.3390/s20226452

**Published:** 2020-11-12

**Authors:** Aldo-Francisco Contreras-González, Manuel Ferre, Miguel Ángel Sánchez-Urán, Francisco Javier Sáez-Sáez, Fernando Blaya Haro

**Affiliations:** 1Centro de Automática y Robótica (CAR) UPM-CSIC, ETS Ingenieros Industriales, Universidad Politécnica de Madrid, Calle de José Gutiérrez Abascal, 2, 28006 Madrid, Spain; af.contreras@alumnos.upm.es (A.-F.C.-G.); m.ferre@upm.es (M.F.); franciscojavier.saezs@upm.es (F.J.S.-S.); 2ETS Ingeniería y Diseño Industrial, Universidad Politécnica de Madrid, Ronda de Valencia, 3, 28012 Madrid, Spain; fernando.blaya@upm.es

**Keywords:** motion capture, soft angular displacement sensors, upper limb, motion tracking, wearable sensors

## Abstract

Motion tracking techniques have been extensively studied in recent years. However, capturing movements of the upper limbs is a challenging task. This document presents the estimation of arm orientation and elbow and wrist position using wearable flexible sensors (WFSs). A study was developed to obtain the highest range of motion (ROM) of the shoulder with as few sensors as possible, and a method for estimating arm length and a calibration procedure was proposed. Performance was verified by comparing measurement of the shoulder joint angles obtained from commercial two-axis soft angular displacement sensors (sADS) from Bend Labs and from the ground truth system (GTS) OptiTrack. The global root-mean-square error (RMSE) for the shoulder angle is 2.93 degrees and 37.5 mm for the position estimation of the wrist in cyclical movements; this measure of RMSE was improved to 13.6 mm by implementing a gesture classifier.

## 1. Introduction

In the fields of biomechanics, physiotherapy and kinesiology, motion capture plays a fundamental role in the study of the measurement of physical activity, either to know the performance of a device or the evolution of a therapy. Optical motion capture is one of the most accurate methods for measurement of human kinematics [1]. In addition to the fact that the use of this technology is expensive, it limits mobility to specific areas and, in most cases, bulky markers and special suits must be carried by the user. Outside of a controlled environment, a common solution is the use of inertial measurements units (IMUs), which in some cases are small and wireless, and are capable of obtaining sample measurements at high speeds [2]. These sensors may be suitable for short periods of time where the accumulated error is not significant. Nevertheless, a high demand of hardware resources is used for data processing as filtering, integration and trigonometry operations are involved to estimate joint angular kinematics [3,4]. Users should also consider movement complexity, sensor placement, the studied joint, biomechanical models used and calibration procedure.

The era of flexible systems is on the rise, and a new generation of motion sensors is emerging. The wearable flexible sensors (WFSs) are created with flexible materials which are inexpensive to manufacture, have better mechanical and thermal properties than non-flexible sensors and are lightweight and comfortable for motion capture [5]. Still, there are important features to consider, such as stretching, compression, bending and twisting [6]. With the use of wearable systems, there is a promising path to contribute to a meaningful diagnosis of shoulder conditions, as well as a concise follow-up for rehabilitation evolution [7]. Upper extremities of the human body have complex kinematics and contain a large number of degrees of freedom (DOF) [8]; the glenohumeral (shoulder) joint, for instance, is a complex joint with more than three DOF [9]. WFSs could properly estimate the kinematics of the shoulder due to its physical characteristics by eliminating the use of rigid parts and even communication cables in some cases [10,11,12,13]. It has been proved that it is possible to estimate the angles of the arm with respect to the trunk by placing sensors on the body [14,15,16,17]. Additionally, the use of compression jackets, combined with soft sensor arrays, makes it possible to accurately estimate the position of the limb [18,19], as well as to place the sensors onto the skin without the use of adhesives or other preparations such as invasive intrusions in the user’s body. However, a disadvantage of compression garments is that sweat can cause damage when it comes into contact with electronic devices.

Work-related studies using IMU systems for shoulder orientation estimation reported a RMSE of 15${}^{\circ }$ for flexion/extension movements, a RMSE less than 20${}^{\circ }$ for abduction/adduction movement and from 1 to 60${}^{\circ }$ for rotation movement [20]. The combination of IMU with electromyography sensors (EMGs) shows a RMSE of 10.24${}^{\circ }$ in complex tasks performed with the shoulder [21]. Another study using an IMU system as reference and a smart-textile with printed strain sensors showed a mean error of 9.6${}^{\circ }$ in planar motions measurements of shoulder joint [22]. In the case of lower limbs using WFSs, the maximum RMS error was nearly 15${}^{\circ }$ for the knee sensors [23]. The authors of [24] developed a dynamic measurement system for the movement of the legs, which can detect the squat position with an accuracy of 3${}^{\circ }$ and walking with an accuracy of 5${}^{\circ }$. WFSs made of elastomers were used for sensing hand gestures, obtaining an error of 5${}^{\circ }$ [25], which is acceptable to the majority of requirements of motion capture.

In robotics, the end-effector position is calculated using kinematics equations by knowing the angles of each joint and the length of each link; this means that, to estimate the position of the wrist, the angles of the arm and forearm and its lengths must be known. In this work, the goal is to sense four DOF in the arm [26,27], three DOF in the shoulder and one in the elbow. In addition, a kinematic algorithm is proposed to find the position of the elbow and wrist. These, using mainly the signals from the WFSs and for one single DOF, constitute the IMU system. This document describes the method to recreate a WFS to obtain orientation and position for the upper limbs, so forth applied to the human right arm. A calibration algorithm to estimate the length of the arm and forearm of each subject and an algorithm for the estimation of the position of the elbow and the wrist from the shoulder are proposed. In Section 3, the placement of the sensors and the required calibration for each user are described. Fusion data methods and the filters applied to the resulting signals are discussed. The device presented here eliminates the rigid elements that might interfere in daily tasks and movements, allowing the user freedom and mobility. Furthermore, compared with previous work by the authors, described in Section 2, the number of sensors was reduced, and the computation cost was improved.

## 2. Previous Work by the Authors

It was proved that it is possible to obtain 95% of the variance of the main components for shoulder gestures with an array of seven single-axis resistive sensors [19]. Other results showed that it is possible to estimate the gestures of the shoulder with a performance 95.4% using an array of four WFSs and EMG signals [28]. In this work, a rigorous extension was performed using the two-axis sADS whose operation is explained and detailed in [29,30]. Features of the sADS include linear and bidirectional response, measurement of angular displacement in two orthogonal planes with a sensitivity of 0.016${}^{\circ }$ and a repeatability of 0.18${}^{\circ }$. The capacitive sensors are made using layered medical grade silicone elastomers doped with conductive and nonconductive fillers.

Initially, four sADS were placed in the intermediate positions of the seven single-axis resistive sensors configuration recommended by [19]. Replicating the proposed 20-layer hidden neural network method, with a configuration for the acquired data from the four sADS of [70%, 15%, 15%] for training, cross-validation and test stages, respectively, an overfitting was identified. In order to lessen the chance of overfitting, a principal component analysis (PCA) was performed; the sensor placed over the middle deltoid muscle was found to contain the most activity on pure abduction and adduction movements, while the sensors at the rear, placed over the posterior deltoid, were representative for horizontal adduction and flexion and extension movements. A 92% representation of the ROM was obtained using only two sADS; the study and location of each sensor is detailed in the following subsection.

### sADS Array Location

Three different arrangements were made in different locations at the shoulder as shown in Figure 1. The arrangement resulting from the analyses was the placement of two rotated sADS 90${}^{\circ }$ between two planes that represent 92% of the variance of the data (See Table 1).

## 3. System Overview

This study was performed on a single healthy limb without reduced mobility, the right arm. However, it can be replicated for both arms. The areas considered in the estimation started from the centre of the head of the humerus to the centre of the junction of the radius and the ulna at the wrist. The supination and pronation gestures were not considered.

### 3.1. Working Range

The arm at rest was considered as the natural pose, making a full extension of the elbow as shown in Figure 2a.

The zero-position of the system occurred when a pure shoulder abduction gesture reaching 90${}^{\circ }$ (without performing any flexion or extension of the shoulder) and a full extension of the elbow were performed, starting from the natural pose.

### 3.2. Sensors Placement

Placing sensors on a compression jacket presents various difficulties, the most important being wrinkles and folds in the fabric and skin tightening in specific areas. To eliminate such problems, two small rigid parts were designed to pose and guide the two-axis sADS of BendLabs [30]. One of them holds the sensor in a fixed location and the other allows it to slide inside it, not only eliminating the problem of skin stretching but also creating a straight line formed by the limb, which allows the accurate measurement of the angle (see Appendix A.1). The pieces are sewn on the shirt and do not impede the free mobility of the user in any way.

As a result of the PCA for the location of the sADS on the shoulder, the best arrangement was selected with respect to the number of sensors and the representation of movements performed. It is worth mentioning that the locations of each sensor were initially placed in the way recommended by the previous study. The adjustments made to the final positions were identified when making several data captures for subsequent analysis. The location description is detailed on Appendix A.2.

The IMUs were placed on opposite sides of the arm (Figure 3). IMU${}_{1}$ was located just where the shoulder ends and IMU${}_{2}$ was located just before the beginning of the elbow. The orientation of the IMUs should be towards the same direction of the arm.

### 3.3. Shoulder Rotation

As the study presented in this document was implemented in an Exosuit developed in our group [31], these solutions were discarded for the first two DOF (abduction/adduction and flexion/extension). The third DOF of the shoulder (Figure 2e medial/lateral rotation) is revealed at the forearm and can be measured at the elbow. Given that the shoulder sADS array does not detect the rotation gesture, the use of IMU is proposed exclusively for this gesture.

Two IMUs were placed on the arm (same link) as shown in Figure 3. The rotation was calculated as the difference of the rotation angles on the *x*-axis of the IMU. This angle was generated by the shoulder rotation, thus completing the variables necessary to estimate the elbow and wrist position $\delta_{1}$.
(1)$$\delta_{1}=\Delta \left(IMU1_{yaw},IMU2_{yaw}\right)$$

### 3.4. Ground Truth

To verify the estimation proposed in this article, the OptiTrack system was used for ground truth data. A total of 10 reflective pointers and four cameras from the OptiTrack system [32] were placed, providing submillimetre precision. The markers placement generated straight lines between each section to be analysed. Two markers were placed on the back to know the inclination of the user—two on the shoulder (just above the fixed parts of the shoulder sADS), two on the arm, two on the forearm, one on the elbow and two on the wrist [33], as shown in Figure 3.

### 3.5. Data Acquisition

The three sADS were connected to a custom acquisition board based on the LAUNCHXL-F28379D development board and communicated with the micro-controller via I2C protocol at a frequency of 200 Hz. The IMUs and the custom acquisition board were connected to a NVIDIA Jetson Nano via LpBUS and SPI at 400 Hz and 500 Hz, respectively (Figure 4). The LPMS-URS2 9-axis IMU manufactured by LP-RESEARCH, with a Kalman processing stage on board and gyroscope, accelerometer and magnetometer, was selected. This sensor delivers filtered data as output at frequencies up to 400 Hz with a resolution of 0.01. Software was developed on the Jetson Nano board to trigger sensors and store all sensor data along with a timestamp.

To perform data collection for session testing, the compression jacket with sADS and IMUS was placed on the subject. Then, OptiTrack markers were located as shown in Figure 3 in order to obtain the real position of the subject’s arm. Both systems started the data capture operation almost at the same time, first in the OptiTrack software and then in the Jetson Nano. The Jetson Nano stored the following in a file (with the start time in milliseconds): the six angles received from the sADS, the Euler angles, the quaternions and a timestamp of each of the two IMUs and the gesture performed in that timestamp, in addition to a timestamp generated by the Jetson Nano itself. The OptiTrack software stored the points in coordinates *x*, *y* and *z* in the space of each reflective pointer, whose name is the start time in milliseconds, in a file. To synchronise the data of the files of each session (Appendix A.3), the start time and the sampling frequency in each file were considered.

### 3.6. sADS Behaviour

The angular displacement of the sADS was obtained by a differential capacitance measurement. The sensor repeatability, according to the manufacturer, is 0.18${}^{\circ }$. However, when taking measures from the sensor at 200 Hz, a noise of a maximum amplitude near 3${}^{\circ }$ was observed. A set of sensor samples obtained while bending the sensor was stored and a frequency domain analysis was realised over those samples. Following frequency spectrum analysis, a low-pass first order infinite impulse response (IIR) digital filter with a cut-off frequency of 10 Hz was applied to the raw data provided by the sensor to reduce noise. Finally, a $\pm 0.15^{\circ }$ dead zone for sampling data was added to the filter (see Figure 5).

## 4. Estimation Methods

### 4.1. Angle Fusion Algorithm

The fusion of the sensor signals is done by using the four angles given by two sADS sensors. Sensor positions and definition of planes are described in Section 3.1. To find the angle in the XY plane, the xy upper sADS and xz back sADS angles are used: for the XZ plane, the xz upper sADS and xy back sADS angles are needed (see Table 2). Data fusion algorithm is detailed on Appendix B.1.

### 4.2. Method to Estimate Arm and Forearm Length

An important error for the calculation of the position in space of the wrist and elbow with respect to the shoulder—the use of a measuring tape to determine the length of each link (length of the arm and forearm)—was detected. In order to reduce this error, an L-shaped structure was proposed. This calibration method consists of a plane marked by four points at known distances to estimate arm and forearm lengths by trigonometric methods.

This structure was created with the purpose of marking a fixed distance from the shoulder to a point in space, and it was arranged in four points: two fixed and two movable; over each one of the four points, there was a vertical pin with a magnet on top of it, which paired with a bracelet for the user that locates the magnet in the centre of the wrist. By performing a series of defined movements (Appendix B.2), the ADS sensors were calibrated and the arm length was calculated with trigonometric equations.

### 4.3. Elbow and Wrist Position Estimation

To estimate the position of the shoulder and the wrist, it is necessary to know the angles $\alpha_{1}$ (Plane XY), $\phi_{1}$ (Plane XZ), the rotation of the shoulder $\delta_{1}$ (Equation (Equation 1)) and the elbow flexion angle $\alpha_{2}$, as well as the distances of each segment: shoulder to elbow (Equation (Equation 17) $arm_{length}$) and elbow to wrist (Equation (Equation 18) $forearm_{length}$). The kinematic model considered in this work consists of 4 rotating joints. The first three joints take into account the glenohumeral joint. It is commonly accepted that said articulation contains two translational DOF. However, since the surfaces of the head of the humerus are more than 99% spherical [34], it is modelled as a spherical joint. The fourth joint represents the elbow, where pronation and supination are neglected. Finally, to estimate the position of the arm and elbow, the Denavit–Hartenberg notation is used (Appendix B.4).

## 5. Experiment

The experiment depended on two scenarios. The first one consisted of a table, a chair and the structure described in Section 4.2 in order to perform the length estimation (system calibration). The second one was needed to acquire the positions with the ground truth system (GTS) and to perform the analysis of acquired data (ground truth pose). A graphical user interface developed on the Jetson Nano board allowed calibrations to be performed by saving the values for each user and the gesture performed and guiding the speed and kind of movement of the participants while performing the gestures. Once the estimates of arm and forearm length were saved, these data were used for position estimation. In this document, all the acquired data that were stored to be analysed by the MATLAB R2020a software tool.

The study involved nine subjects: two females and seven males for both scenarios. The subjects were aged 27 ± 4.4 years (mean ± standard deviation (SD)), with a body weight of 81.7 ± 15.6 kg and a height of 1.83 ± 0.16 m. All participants had no evidence or known history of skeletal and neurological diseases, and they exhibited normal joint ROM and muscle strength. All experimental procedures were carried out in accordance with the Declaration of Helsinki on research involving human subjects and approved by the ethical committee of the Polytechnic University of Madrid.

### 5.1. System Calibration

The arm calibration algorithm (Section 4.2) was tested with nine subjects, measuring the arm length (from shoulder flexion point to elbow flexion point) and the forearm length (from the point of flexion of the elbow to the centre of the point of flexion of the wrist) with a tape measure and placing the reflectors of the OptiTrack ground truth system (GTS) at each point (see Table 3).

The angles were captured by the interface, which displayed them on screen along with the time of capture and the estimation of arm and forearm length, obtained using the Equations (Equation 15) and (Equation 16) described in Appendix B.2 with a distance $\bar{AB}=39.5$ cm. The calibration frame is shown in Figure 6. Then, the distances estimated were compared with the GTS.

The performance of the frame developed is shown on Table 4 and presents an RMSE of [0.540, 2.280 mm] (best, worse case) for the arm and RMSE of [0.200, 3.130 mm] for the forearm.

### 5.2. Ground Truth Pose

The tests were carried out in four different sessions and the data acquired in the tests were analysed later. The first session consisted of a sequence of five abductions/adductions, five flexions/extensions and five horizontal abductions. This session was performed with visual feedback through a graphical interface projected on a monitor, where the time of each gesture (e.g., flexion) was four seconds with one second of rest—that is, each complete movement (e.g., flexion + extension) had a duration of ten seconds. In the second session, free movements were performed for 30 s through the lower shoulder area—that is, only obtaining negative values according to the range of work shown in Figure 2. The third session consisted of making free movements in the upper area. In the fourth session, free movements were performed throughout the entire working range. The subject performed each whole session five times.

The shoulder, elbow and wrist markers were used to make a comparison of the position of the GTS with the estimation made by the proposed algorithm. The back, arm, forearm and shoulder markers were used to find the angles generated by the poses. The shoulder marker was placed on a 12 mm raised pin on the *z*-axis; therefore, this elevation was subtracted to make the comparison, this being the reference point (P) of the system ($X_{0}$, $Y_{0}$, $Z_{0}$). The pose of the other markers was defined by subtracting the reference point from the position of each marker given by the GTS—that is, $x_{n}=x_{n_{GT}}-x_{0}$.

### 5.3. Orientation Method

The performance of the orientation estimation was evaluated offline using the RMSE for two aspects: the angles acquired by the sADS and the position estimation of the elbow and wrist compared to the GTS, obtained with the L-shaped structure (Figure 7). The estimation of the elbow and shoulder position depended on the angles measured by the sADS in the experiments. The position of the GTS markers was used to find the error of the angles. The angle between the markers of the OptiTrack system was obtained by calculating the angle generated between markers on the arm (placed on the shoulder acromion bone) and the ones on the back. The data from the experiments were classified into cyclical movements, free movements, movements under the shoulder, movements over the shoulder and each gesture performed. The angles acquired by the different sensors and the angles generated by the location of the markers were compared.

It can be seen in Table 5 that free movements and movements carried out over the shoulder had a greater error than cyclical movements and those carried out below the shoulder. In addition, it was observed that the sensor placed on the back measured the gestures of flexion/extension and abduction/adduction more easily, while the sensor placed on the top was a better solution for horizontal adduction movements and shoulder movements.

Using the algorithm proposed in the Section 4.3 and the angles captured from the elbow and wrist experiments, it was possible to compare the positions with the GTS. The largest error obtained in the entire experiment was 124.3 mm, which corresponded to the position of the wrist in a free movement exercise on the shoulder, and the smallest error was 36 mm, which corresponded to the elbow in a cyclical movement below the shoulder. Table 6 shows the RMSE of all captured movements classified in the elbow and wrist in free and cyclical movements, above and below the shoulder.

The mean error of the position estimate was 43.4 mm for the elbow and 57.2 mm for the wrist in all three axes. If the length of the arm of an average adult is ≈300 mm, the error would be equivalent to 9.53% of the length from shoulder to the elbow, or 14.47% from the shoulder to the wrist and, in the worst case up to 27%. This means that, when extending the arm entirely, it can be said that the error is as large as half of the palm of an adult’s hand in space.

To improve performance in cyclical movements, a classifier was developed; the classifier identifies the gestures performed and also identifies the movements that correspond to the lower and upper area with respect to the shoulder. The classifier was generated by making a comparison with 24 different methods (see Table A2 in Appendix C.1). Once the best method was obtained and trained, a function was created in MATLAB 2020a using the upper sADS and back sADS angles to classify the motion.

After the classification algorithm found which gesture the movement belonged to, the method described in Appendix C.2 was used, applying a different “weight” for each case, through Equation (Equation 12). The weights (Table 7) were found through an empirical method using the data of the participants of the first test.

By evaluating the fusion algorithm with the gesture classifier, an overall improvement in angle measurement was observed. This algorithm used the sensor signal that performed best (upper or back sensor) for each type of gesture, resulting in the smallest measurement error of the shoulder angles. The RMSE for cyclical movements was 2.93${}^{\circ }$. In the case of free movements, the algorithm was allowed to interpret each signal from the sADS sensors and classify the movements, obtaining a significant improvement in angles with an RMSE of 3.71${}^{\circ }$. Because the mean angle error decreased, the position estimate improved (Table 8). It should be mentioned that the largest error obtained for the position estimation using the weighted fusion algorithm was 60.7 mm, 43% less than the largest error without the weighting. The smallest was 12 mm, only 19 mm better than the smallest without the weighting.

The overall RMSE was 26.7 mm. In case the movements were performed below the shoulder or cyclically (as is the case in rehabilitation processes), the error decreased. The horizontal adduction gesture generated a greater error in the estimation, since it was carried out in the limit of the movements that were classified in the upper and lower part of the shoulder. Furthermore, this gesture also shared zones of movement with all the other gestures.

## 6. Discussion

During the calibration process, it was difficult to correctly place the frame without the appropriate measurement system for the angles generated by the shoulder. It was identified that initial position calibration can be improved using structures fixed to the ground or to the user’s hip. With an abduction of more than 90${}^{\circ }$, a scapular movement was generated, causing a displacement of the superior sADS and, consequently, errors in the measurement. In experiments of over 30 min with the users using the shirt, no evident change in the behaviour of the sADS signals was observed due to prolonged periods of time; the IMUs were re-calibrated in each data collection. In this work, IMUs were used only for the shoulder rotation gesture. For defined tasks, the sensing of shoulder rotation could have been eliminated, removing the use of IMUs and facilitating donning-doffing and estimation calculations. This work will be applied on an exosuit and integrated into an online system; it may use a different classification method than used on this study due to the time response, and a robustness investigation is needed.

The maximum calculated error of the calibration of the OptiTrack cameras was 0.343 mm and can be considered in the comparison of the method of estimating the length of the arm and forearm, being able to improve or worsen within that range. If the noise from the sADS is not reduced, a projected noise is produced at the wrist. Similarly, not using the arm and forearm distances estimated by the L-shaped structure method increases the error. These behaviours increase the error up to 3.5${}^{\circ }$ and are projected in the final position (wrist), increasing up to ≈ 50 mm.

Our results are similar to [35], where, on a cyclic movement, there is an RMSE of [1.42${}^{\circ }$, 3.89${}^{\circ }$] (best, worst) and, on free or random movements [1.88${}^{\circ }$, 4.41${}^{\circ }$]. We also searched for the optimal sensor arrangement, finding four essential sensors for the estimation of the orientation. On the contrary, we found that two sensors can represent 92% of the ROM without adding scapular movement. Similar studies used piezoresistive strain sensors directly adhered to the skin to estimate shoulder ROM; the comparison between reference data from OptiTrack and the strain sensors showed a RMSE less than 10${}^{\circ }$ in shoulder flexion/extension and abduction/adduction estimation [36]. In the case of [37], which used a smart compression garment, the best result for angle measurement of the elbow with 9.98${}^{\circ }$ of error was obtained. An interesting study using 20 self-made soft strain sensors [38] showed an overall error position of the full-body motion tracking of [29.5 mm, 38.7 mm] (best, worst case) and results for the shoulder joint with just two sensors of [30.9 mm, 47.1 mm] for the elbow [13.5 mm and 34.7 mm] and for the wrist [27.3 and 43.5] using OptiTrack as ground truth.

## 7. Conclusions

Wearable flexible sensors are becoming common as a technology used for motion capture. The shoulder is a complex joint and plays an important role in daily living activities. Therefore, designing portable and accurate arm motion capture system can be challenging. This document reports a device capable of obtaining a mean RMSE of 3.54${}^{\circ }$ in the angle measurement of the shoulder. Using the angles and the length of the arm and forearm and compared with a GTS, a RMSE of 29.1 mm was found for the estimation of the position of the wrist. This performance may be useful in applications of physiotherapy, kinesiology or gaming. However, due to the lack of accuracy at a millimetre level, the use of this device for robotic or biomechanical systems could limit its approaches. Still, a great possibility can be exploited in different applications that require monitoring of the human body in open spaces.

## Figures and Tables

**Figure 1 sensors-20-06452-f001:**

Different locations for the variance estimation. (**a**) Four soft angular displacement sensors (sADS) were placed in the recommended intermediate poses. (**b**) The frontal sADS was eliminated due to its low representation in the variance; it also generated obstructions and poor measurements in the horizontal gestures. The three previous poses were preserved. (**c**) Placement of two sADS, one in the centre of the two sADS of the top three sensor array, and the other in the rear, generating an angle of $90^{\circ }$ between the two planes with respect to the first.

**Figure 2 sensors-20-06452-f002:**
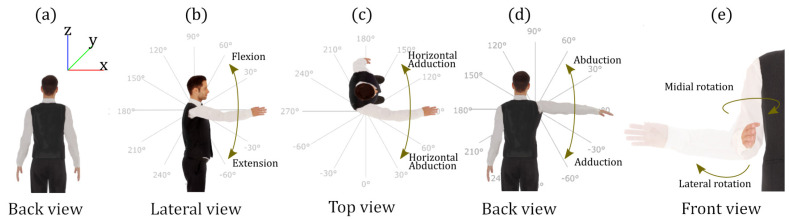
Gestures that define the work space of the right arm. (**a**) Rest position. (**b**) Flexion/extension gesture corresponding to the YZ plane. (**c**) Horizontal abduction/adduction corresponding to the plane XY. (**d**) Abduction/adduction gesture corresponding to the XZ plane. (**e**) Depiction of the rotation of the shoulder, named $\delta_{1}$.

**Figure 3 sensors-20-06452-f003:**
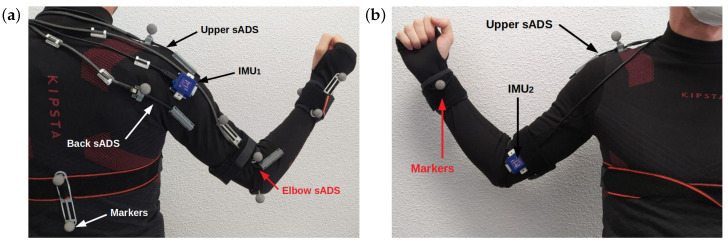
Location of sensors on the compression jacket. (**a**) Back view. The reflective markers are shown in bright grey, the IMU${}_{1}$ and the two sADS on the shoulder (upper and back) and the sADS on the elbow. (**b**) Front view, which shows two reflective markers and the IMU${}_{2}$.

**Figure 4 sensors-20-06452-f004:**
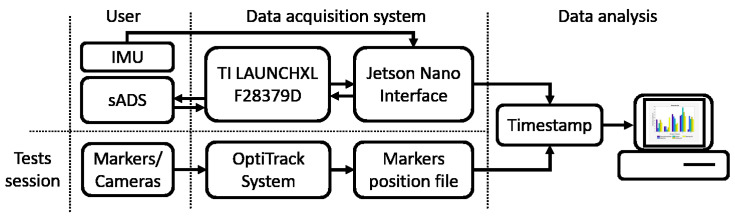
Flow diagram of the data acquisition system. On the left side, the movements of the user, the generated angles and the poses of the markers are shown; in the centre, the acquisition of data and communication between boards as well as the storage of positions of the reflective pointers; on the right side, the data analysis based on the timestamp of each file.

**Figure 5 sensors-20-06452-f005:**
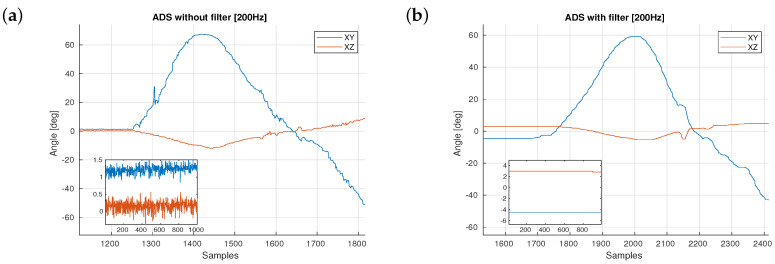
Example of similar gestures in two different takes. (**a**) First take shows the behaviour of the sensor without the filter. (**b**) Second take of a similar gesture shows the behaviour with the filter.

**Figure 6 sensors-20-06452-f006:**
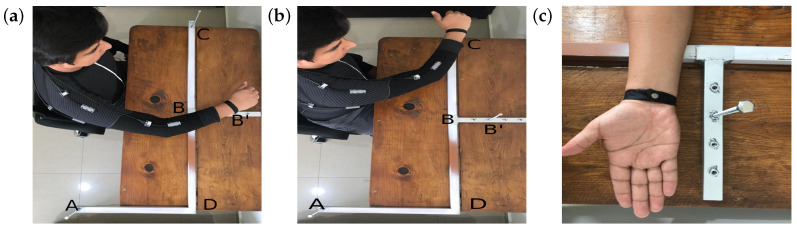
Calibration frame top view. (**a**) Shows arm pose at the point *B* of the frame to estimate the length of the arm and forearm with a known distance $\bar{AB}$. (**b**) Gesture performed for redundant calculation towards point *C*. (**c**) It shows the bracelet tight and a magnet to guarantee the same pose of the wrist with the pin on the frame.

**Figure 7 sensors-20-06452-f007:**
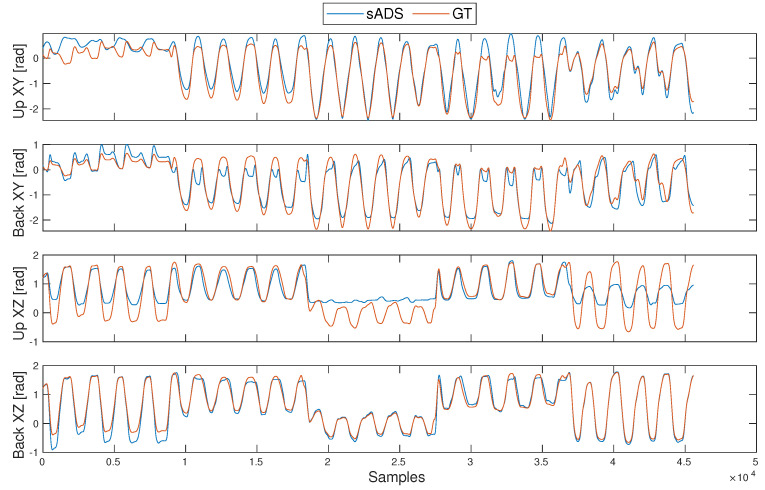
Performance of the sADS angles compared with the GTS in a series test.

**Table 1 sensors-20-06452-t001:** Result of the estimated variance for different sADS arrays.

Type of Sensor	Amount of Sensors	Variance
Single-axis resistive	7	95%
Two-axis capacitive	4	93%
Two-axis capacitive	3	92%
Two-axis capacitive	2	92%

**Table 2 sensors-20-06452-t002:** Merged data is obtained in two different fusions using the shoulder ADS sensors.

Fusion	Upper sADS	Back sADS
Plane XY ($\alpha_{1}$)	xy-axis $\to \theta_{1}$	xz-axis $\to \theta_{2}$
Plane XZ ($\phi_{1}$)	xz-axis $\to \theta_{1}$	xy-axis $\to \theta_{2}$

**Table 3 sensors-20-06452-t003:** Comparison between the estimation made by the proposed algorithm with the ground truth system (GTS) and the tape measure of the arm and forearm lengths of the subjects (mm).

	Arm				Forearm		
Subject	Tape	Estimated	GTS		Tape	Estimated	GTS
1	300.0	**311.52**	**310.98**		250.0	246.20	246.83
2	330.0	331.40	332.08		270.0	260.44	261.32
3	345.0	340.78	339.12		300.0	291.83	290.55
4	310.0	312.86	311.98		260.0	272.08	273.56
5	334.0	332.27	332.98		260.0	258.40	256.56
6	295.0	305.89	304.63		250.0	**249.32**	**249.12**
7	330.0	**328.63**	**326.35**		270.0	277.42	278.77
8	315.0	309.43	308.66		255.0	260.18	260.80
9	340.0	337.82	339.63		295.0	**302.81**	**299.68**

**Table 4 sensors-20-06452-t004:** Performance of the proposed frame (RMSE). Comparison between measurements.

	Tape vs. Estimated	Tape vs. GT	Estimated vs. GT
Arm	5.96 mm	5.87 mm	1.31 mm
Forearm	7.18 mm	7.48 mm	1.50 mm

**Table 5 sensors-20-06452-t005:** Orientation error in root-mean-square error (RMSE) obtained by the ground truth system (GTS) when performing the movements and gestures. Elbow orientation error was compared even in gestures that did not involve elbow movement.

Sensor	Cyclic Mov.	Free Mov.	Below the Shoulder	Over the Shoulder	Flexion/ Extension	Abd./ Add.	Horizontal Adduction	Rotation
Upper sADS	$6.84^{\circ }$	$7.33^{\circ }$	$1.52^{\circ }$	$4.18^{\circ }$	$10.53^{\circ }$	$4.22^{\circ }$	$3.71^{\circ }$	n/a
Back sADS	$4.50^{\circ }$	$5.67^{\circ }$	$2.01^{\circ }$	$8.39^{\circ }$	$2.97^{\circ }$	$1.93^{\circ }$	$9.66^{\circ }$	n/a
IMU	$1.93^{\circ }$	$2.86^{\circ }$	$2.96^{\circ }$	$1.66^{\circ }$	$2.30^{\circ }$	$1.89^{\circ }$	$2.23^{\circ }$	$1.51^{\circ }$
Elbow sADS	$0.43^{\circ }$	$0.93^{\circ }$	$1.73^{\circ }$	$1.53^{\circ }$	$0.54^{\circ }$	$0.38^{\circ }$	$0.78^{\circ }$	n/a
Mean	$3.42^{\circ }$	$4.19^{\circ }$	$2.05^{\circ }$	$3.94^{\circ }$	$4.08^{\circ }$	$2.10^{\circ }$	$4.09^{\circ }$	n/a

**Table 6 sensors-20-06452-t006:** Error for the estimated position by the shoulder sADS given in RMSE (mm).

	Elbow Position				Wrist Position				
	Cyclic Mov.	Free Mov.	Below the Shoulder	Over the Shoulder		Cyclic Mov.	Free Mov.	Below Shoulder	Over the Shoulder
Upper sADS	39.7	**61.5**	31.7	46.5		**59.3**	**79.2**	**58.0**	49.7
Back sADS	35.4	43.8	30.3	**58.6**		36.3	**48.8**	44.3	**82.2**

**Table 7 sensors-20-06452-t007:** Weights used in Equation (Equation 12) to reduce the estimation error.

	Below Shoulder	Over the Shoulder	Flexion/ Extension	Abduction/ Adduction	Horizontal Adduction
Upper sADS XY → aFactor	0.4	2.5	2.3	1.6	2.3
Back sADS XZ → bFactor	2.2	1.7	1.9	2.5	1.0
Upper sADS XZ → aFactor	2.0	2.5	0.4	2.5	1.2
Back sADS XZ → bFactor	2.2	2.3	2.7	1.8	2.7

**Table 8 sensors-20-06452-t008:** Fusion RMSE for position estimation (mm).

	Cyclic Mov.	Free Mov.	Below the Shoulder	Over the Shoulder
Elbow	**12.3**	31.6	14.1	39.5
Wrist	14.9	**40.9**	20.3	40.4

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
