# Peer review of "Efficient Upper Limb Position Estimation Based on Angular Displacement Sensors for Wearable Devices"

_sensors, 2020, doi:10.3390/s20226452_

Round 1

Reviewer 1 Report

Thanks to the authors for the great effort made to improve the quality of their research. All my concerns and suggestions have been addressed properly.

Extensive changes have been made both in methodology, validation, increase of number of  subjects and comparison with the state of the art. Overall quality has been improved significantly and my recommendation is “Accept in present form”

Reviewer 2 Report

This paper presents the development and evaluation of a flexible wearable device for measuring upper limb pose. The work is somewhat interesting and potentially novel, but is crippled by poor presentation that prevents readers from following the development process or gauging the actual originality/novelty/significance relative to previous work. I already reviewed a previous version of the manuscript and believe that the authors have put some effort into improving the manuscript, but that it was not sufficient. Major comments are below.

1. The introduction and previous work need to be rewritten to better place the current paper with regard to the state of the art. The authors mention that "only a few studies" have used wearable flexible sensors, but do not discuss any of these studies in the introduction or previous work; the only paper they cite is a review paper rather than original research. Authors have made some effort to try to contextualize their work in the discussion, but it needs to appear earlier. From the first two sections, the novelty appears to be mainly in extending previous work with single-axis sensors to two-axis sensors.

2. The introduction and previous work are very disjointed, jumping from topic to topic with no clear flow, presenting ideas that never reappear (e.g., complaining about lack of standardization then never mentioning this again), and making it difficult to tell what is done in the current study, what was done in the authors' previous work (which takes up most of the Previous work section, ignoring other authors' work), and what was done in previous unrelated work. I asked the authors in the previous review to please rewrite the text so it flows better and is more easily understood by the reader, but they have not adequately done so.

3. Much of the work is presented in appendices for no clear reason - the appendices include critical aspects of the work like the experiment protocol and the placement of the sensors. The algorithmic details can partially stay in the appendix, but everything else should be in the main text.

4. The authors heavily emphasize the need for wearable flexible sensors that do not impede the user, but then still use IMUs (which they criticize) as a critical part of the system, and use a compression jacket that seems fairly inconvenient for the user. I would encourage them to better discuss the actual user-friendliness.

5. The results mix displacement error in cm/mm, joint angle errors, and classification accuracy, with no clear relation between them. First, it is unclear how displacement in cm/mm was actually obtained - did the authors continuously estimate each sensor's position in 3D space? If so, why? Wearable sensors should primarily be used to measure body variables, not global variables. Second, different types of results are not comparable. For example, authors state (line 276) that IMUs + EMG show RMSE of 10 degrees in complex tasks, then state (line 278) that the authors have "on the other hand" shown that WFS + EMG can classify gestures with a performance of 95.4%. Classification accuracy and joint angle relative to a ground-truth system are completely different things, and cannot be directly compared with "on the other hand" statements. 

6. Many of the figures and tables are vague and difficult to understand. I previously requested in my last review that the authors make the tables more accessible to the casual reader, but they have not done so. For example, it is not even clear if "Global" in tables refers to using all sensors for all variables of interest, using all sensors for some variables, etc. Especially confusingly, some tables appear to imply that, e.g., elbow sensors have been used to estimate shoulder angles, which would make no sense.

7. In the human subjects protocol, the speed of the movements is not given even though it has a major effect on measurement accuracy.

8. The authors use what appears to be an L-shaped mechanical structure to measure arm length, but it is unclear why this is actually necessary. It is doubly confusing since the authors appear to compare the results of this structure to results obtained with measuring tape as ground truth; if the measuring tape is ground truth, just use that.

9. The crossvalidation of the classifier is not fully convincing. Authors compared 24 different classification methods, and stated that they performed 5-fold crossvalidation. However, did they just obtain crossvalidation results for all 24 methods and then took the best result of the 24? If so, this is an inappropriate approach - the authors should perform an inner/outer crossvalidation where, e.g., all 24 methods are trained on 60% of the dataset, tested on 20% of the dataset, the best method is selected, and then the best method is tested again (unchanged) on the remaining 20% of the dataset, with that being the final accuracy.

Reviewer 3 Report

The paper is focused on the design and experimental verification of a system for motion capture of the shoulder. It can be agreed with the authors that wearable flexible sensors are becoming common in this application and the shoulder being a complex joint plays an important role in everyday life. The paper says the instrument provides relatively good accuracy i.e. a global RMSE equals 3.54 degrees, which appears to be better than in the case of previous systems cited by the authors. Obviously, this is not a level suitable for robotics though it is precise enough for physiotherapy and similar issues.

The paper as a whole is well written, easy to follow with very good command of English language. The figures and plots are of high quality and clearly described. The authors have cared of details and precise wording. It is a good idea to add some useful information in a form of appendices, as they provide additional insight to the issue in question, particularly for readers seeking more profound analysis.

I think that the paper can be published in its current form and can be helpful for researchers working in the field.

Round 2

Reviewer 2 Report

The authors have improved the work, and I now believe it should eventually be published, though I would suggest another round of revisions since not all points were adequately addressed. Remaining issues:

1. While the authors have done a better job of describing the state of the art in the introduction, the specific gap and novelty are still unclear in the introduction. The authors state that WFS made of elastomers used for sensing hand gestures [25] have an error of 5 degrees, which is acceptable to the majority of requirements of motion capture. Then why do we need additional work? And what is the novelty relative to previous work besides just using 2D vs 1D devices?

2. I still do not understand the point of the L-shaped structure used to measure arm length if measuring tape is the reference. Are there any situations in which this structure would be ready available and researchers would not have access to measuring tape? If not, what is the point of replacing reliable and convenient tape with a more expensive device?

3. I did not understand the explanation of position estimation; I still do not fully understand what it refers to or what position error refers to. I also do not understand why the authors chose to standardize to mm when the base measurements seem to be in angles. Why not just standardize to angles which make more sense for joints?

4. I do not understand why there is a "Mean" row in tables 6 and 8 when the tables only have two other rows and the mean is thus a mean of two numbers.

5. I still do not understand what Table 7 represents.

Author Response

This manuscript is a resubmission of an earlier submission. The following is a list of the peer review reports and author responses from that submission.

Round 1

Reviewer 1 Report

This paper presents a flexible wearable device for measuring shoulder angles and the angle of the elbow. A gesture classifier with a signal fusion algorithm is designed to improve the estimation of the arm pose. Experimental results validate the performance of the proposed pose estimation system. There are some English and math errors in the paper; for example,

  1. Line 60 at Page 2, the sentence misses a subject.
  2. Equation (1) at Page 6, it should be pi/2-a1
  3. Equation (7) and (8) at Page 7, A1 to A5 are undefined.
  4. Line 236 at Page 9, the format of subheading should be corrected.
  5. Equation (A7) at Page 15, the symbol * should be removed.
  6. Line 338 at Page 15, the symbol y should be removed.

The authors should carefully revise the paper before publication.

Reviewer 2 Report

The article presents several serious flaws in both objectives and methodology. There is no comparison with other state-of-the-art methods to obtain the angles and lengths of the limbs. The end effector pose RMSE of 4.07cm in robotics is quite high, being not able to perform effectively pick and place task. The whole process of calculating angles and lengths is carried out off-line without any type of temporal cost analysis when one of the main benefits of using this kind of sensor (IMUs) is obtaining high-frequency results. The proposed method for measuring limb lengths requires additional hardware (l-shaped structured) and is compared to tape measure measurements. The benefits of using this complex system versus just measuring are not justified. In addition, why the available Optitrack system has not been used for more accurate measurements? However, the worst flaw that forces me not to recommend its publication is the sample size. The experiments have only been carried out with four subjects and a single subject. This means that the results obtained are not statistically significant nor can it be extrapolated. The sample size should be at least 20 subjects.

Reviewer 3 Report

Manuscript ID: sensors-884735

Title: Accurate Upper Limb Pose Estimation Based On Flexible Wearable Device

Authors: Aldo F. Contreras-González, Manuel Ferre, Francisco Javier Sáez-Sáez and Miguel

Ángel Sánchez-Urán

Journal: Sensors

The authors describe a wearable device with soft angular displacement sensors and internal measurement units to capture the pose of the arm and forearm and the position of the wrist. In experiments the wearable device was tested on four humans and the angles and positions estimated and compared to the values obtained by the OptiTrack system.  

While reading I came up with a few comments and questions about the manuscript, which are discussed below.

Comments:

C1: In the Abstract and Introduction the authors describe the wearable soft sensor device sometimes as a soft robot. This description is misleading since the device passively measures displacements and angles but does not actively do anything. Also descriptions as wearable jacket, exosuit and compression shirt are used. I recommend to just use one single description, as wearable soft sensor device or similar.

C2: Figure 1. Do the colors of the arrows have a meaning? I was wondering if the color of the arrows indicate the direction of movement but the green arrow in the back view doesn’t fit to my idea. If the colors do not correspond the axis, I recommend to use different colors, or otherwise explain the relation.

C3: The rotation angle delta_1 of the shoulder is not shown in any figure of the manuscript.

To make it easier to understand the used angles, I recommend to introduce all angles and important points (e.g. on page 9: reference point P and X0, Y0, Z0) in a figure, e.g. in Figure 1 or 2. Also adding alpha_1/2, as well as the XY and XZ plane and the angles theta_1/2 in a figure would make it easier to understand how the angles are defined (Table 1).

C4: Table 3 shows all important results of the measured arm lengths and angles with the soft sensor device. I would add the percentage error of the measured arm length to the table to better understand the quality of the measurement

C5: p. 11, line 267-281: The authors describe the error and improvement of the measurements by the fusion algorithm. However, from the text, the impression arises that an RMSE of 4.5° and 8.4cm is not really good. Especially the sentence (line 275) “The smallest is 12 mm, only 19 mm better than the smallest without the weighting” gives me the impression that this is not an accurate pose identification, as the title of the paper suggests. Please compare the quality of your results with other portable sensors and state very clearly if and why your measurement errors are acceptable  

C6: Figure A2: I recommend to enlarge the picture, because the shoulder area is not well visible

C7: Figure A3: Please add point A, B and C to the figure

Typos:

Please check the spelling throughout the manuscript.

  • In the header of section 2.5 you use American English but otherwise you use British English.
  • In the header of section 5 you write pose Estimation instead of Pose estimation
  • sometimes a space is missing between values and units

p2, line 60:          Please check the gramma of this sentence.

p3, line 89:          often the authors use capital letters for the axis direction, however, I recommend to use lower case letters for x-axis, y-axis and z-axis and capital letters for the XY plane etc.

p3, Figure 2:       figure 2a is expressed as back few, hence, I would add front to figure 2b.

p4, Figure 3:       to keep consistent descriptions in the manuscript I would write sADS instead of ADS

p6, enumeration: add the correct figure number to the referenced gesture figure

p6, eq.1:              shouldn’t it be theta_1 = pi/2-alpha_1 instead of pi/2*alpha_1?

p7, equations:    please write cos and sin not italic but use $\cos$ and $\sin$ in Latex

p11, line 273:     in …(Table7)… the second bracket is missing

Reviewer 4 Report

This paper presents the development and evaluation of a flexible wearable device for measuring upper limb pose. While the work has some interesting aspects, I believe that it is crippled by poor presentation that prevents readers from following the development process or gauging the actual originality/novelty/significance relative to previous work. Thus, my opinion is that it should be rejected and that authors should be allowed to resubmit it after a thorough rewrite.Specific major comments below; I have not listed minor comments because of the severity of the major issues. 1. The authors do not contextualize their work with respect to the state of the art. The introduction states that other technologies have been used to measure upper limb pose, then launch directly into a description of their work. There is no description of weaknesses of existing technologies and no description of why their own work is expected to be better. The discussion suffers from the same issue: the authors comment on some issues that came up in their work, but make no effort to compare their results to those obtained with state-of-the-art technologies. 2. Looking at the developed technology in the absence of a literature review (specifically at lines 60-65), it sounds like the authors previously did similar work with single-axis sensors, and the novelty primarily consists of now doing the same thing with two-axis sensors that were already developed in the past. Thus, the novelty does not appear to be very high. Please clarify. 3. The actual research and development is presented in a meandering and unsystematic manner, to the point that it is difficult to understand what was done and why. The manuscript jumps from topic to topic, makes assumptions about understanding of specialized topics, overexplains simple topics, uses a confusing high-level structure that is not helpful to the reader, and offloads critical work to six different appendices. For example, section 3.1 (Angle fusion algorithm) says "data fusion is carried out as mentioned in Table 1", presents a confusing table, and then offloads any other data fusion to the appendix even though this is a critical aspect of the work. Section 3.2 then talks extensively about a "L-shaped structure" to estimate arm and forearm length, but it is unclear if this is a mechanical structure and why a specialized structure is needed to measure arm and forearm length when this could probably just be done with measuring tape. I have not provided detailed comments about all the presentation issues in the manuscript, because there are so many of them. In the end, I think the authors need to do a complete rewrite with the aim of making the manuscript easily understandable for an average Sensors reader. It is not enough to simply fix the above issues of sections 3.1 and 3.2. 4. Figures are simply dropped into the manuscript without explaining what they mean and why we should care. For example, figure 5: Resulting angle from fusion of two sADS. I do not understand what I'm looking at here or why. Similarly for figure 4, why should I care about the difference between a filtered and unfiltered signal, especially since the filter seems to be a basic lowpass filter whose performance is predictable? 5. Similar issue for tables. Table 3: I understand what subject arm and forearm lengths are, but what are the measured angles and estimated lengths? Table 4: What are these sensors? What is "Upper"? Why do some of them seem to refer to positions (e.g., back, elbow) and one just says "IMU"? Table 6: What am I supposed to take away from this? 6. The classifier developed to improve performance (lines 258-262) is not convincing, as it seems like the authors just tested a bunch of different methods and picked the one that gave the best result without performing separate validation or investigating robustness/generalizability. 7. The errors, though the authors do not comment on them, appear unimpressive. For example, on lines 255-257, the authors acknowledge that the average error is about 16.5%, and worst case error is 52.2%. Is this better than the state of the art?
